# Evolutionary Implications of Environmental Toxicant Exposure

**DOI:** 10.3390/biomedicines10123090

**Published:** 2022-12-01

**Authors:** Giorgia Bolognesi, Maria Giulia Bacalini, Chiara Pirazzini, Paolo Garagnani, Cristina Giuliani

**Affiliations:** 1Department of Experimental, Diagnostic and Specialty Medicine (DIMES), University of Bologna, via San Giacomo 12, 40126 Bologna, Italy; 2Laboratory of Molecular Anthropology, Centre for Genome Biology, Department of Biological, Geological and Environmental Sciences, University of Bologna, via Francesco Selmi 3, 40126 Bologna, Italy; 3IRCCS Istituto Delle Scienze Neurologiche di Bologna, via Altura 3, 40139 Bologna, Italy

**Keywords:** epigenetic mechanisms, DNA methylation, contaminants, evolutionary biology

## Abstract

*Homo sapiens* have been exposed to various toxins and harmful compounds that change according to various phases of human evolution. Population genetics studies showed that such exposures lead to adaptive genetic changes; while observing present exposures to different toxicants, the first molecular mechanism that confers plasticity is epigenetic remodeling and, in particular, DNA methylation variation, a molecular mechanism proposed for medium-term adaptation. A large amount of scientific literature from clinical and medical studies revealed the high impact of such exposure on human biology; thus, in this review, we examine and infer the impact that different environmental toxicants may have in shaping human evolution. We first describe how environmental toxicants shape natural human variation in terms of genetic and epigenetic diversity, and then we describe how DNA methylation may influence mutation rate and, thus, genetic variability. We describe the impact of these substances on biological fitness in terms of reproduction and survival, and in conclusion, we focus on their effect on brain evolution and physiology.

## 1. Introduction

Organisms continually adapt to their environments, seeking to optimize fitness in different contexts.

From the earliest emergence of life, all life forms had to interface and adapt to toxic substances derived from the evolving planet, such as oxygen, heavy metals, UV rays, and chemical compounds [1].

Similarly, *Homo sapiens* have always been exposed to various environmental toxins such as particles, smoke from the habitual use of fire, thermal processing food-related toxicants, fecal aerosols, and various pathogens transmitted by domestic animals and favored by high population density that characterized post-Neolithic populations [2]. Even today, some human populations have adapted to their environment through increased frequencies in protective variants against harmful substances. For instance, South American populations showed genetic adaptation to exposure to arsenic (As), a toxic chemical naturally present in nature [3].

However, the spread of the Industrial Revolution and the transformation of technology exposed modern humans to contaminants never faced before in nature, such as products of industrial origin, cigarette smoke, and air pollution [2,4]. These changes are too recent to detect signs of genetic adaptation, and modern clinical practice as well as new build environments further reduce the strength of these selective pressure [5].

However, epigenetic processes have been proposed as important molecular mechanisms for medium-term adaptation, and thus they are likely able to cope with the early stages of exposure to new environmental factors [6,7,8], potentially generating an epigenetic signature of the exposure.

Epigenetic mechanisms, in fact, act by affecting gene expression without directly modifying the DNA sequence [9].

Among the different epigenetic processes, DNA methylation is one of the most studied. 

In mammals, DNA methylation occurs at the level of CpG sites and consists of the addition of a methyl group (-CH3) to the 5′ position of cytosine, resulting in the formation of 5-methylcytosine (5mC). The methyl group addition reaction is catalyzed by DNA methyltransferases (DNMTs), a family of enzymes that use S-adenosyl-l-methionine (SAM) as methyl donor and that can catalyze both the de novo methylation (DNMT3A and DNMT3B) and the maintenance of the already existing methylation state during cell division (DNMT1) [10].

The DNA methylation process therefore depends on the genetic background but is also affected by the surrounding environment [7]. There are many studies that deal with how environmental factors, such as toxins, disease, and nutrition, alter methylation patterns causing hypermethylation or hypomethylation of the DNA sequence on a global, gene-specific, or single-site level [6,7].

Given the important role played by epigenetic mechanisms, and in particular by DNA methylation, in the medium-term response to novel environmental factors to which humans are exposed, this review aims to describe the most recent and informative literature available (mainly on humans and mice) to investigate the potential impact of major industrial-era contaminants (external to the human body) on human evolution. We focus on the effects of toxicants on DNA methylation and analyze their implications in terms of evolutionary forces [11], focusing on reproductive fitness in males and females, survival, biodiversity, mutation rate, and brain evolution (Figure 1).

## 2. Toxicants Exposure and Human Biodiversity

Exposure to toxic agents can be a source of natural variation and contribute to shaping human biodiversity. The two main sources of biological variability related to toxicants exposure and described to date are: (I) genetic variability, observable in human populations as a result of adaptive dynamics (long-term adaptation); (II) epigenetic variability, such as altered DNA methylation profiles, observable both at individual and population level and potentially capable of inducing an increase in the mutation rate (medium-term adaptation).

### 2.1. Genetic Variability

It is known from the literature that human populations exhibit genetic variability related to adaptive phenomena. In fact, during their evolution, humans have been exposed to various environmental factors and, over time, has developed a series of genetic responses.

For example, the aridification of North Africa that occurred 7–11 MYA led to the formation of deserts and savannas, resulting in changes in flora and fauna (herds of herbivores) [12] and an increase in pollen, dust, and fecal endotoxins in the air [2,13]. All these environmental changes exposed early hominids to new types of aerosols. In the context of responding to a new exposure, the receptors of innate immunity are central. The Macrophage Receptor with Collagenous Structure (MARCO), for example, is a class A scavenger receptor (SR) that is expressed on the surface of respiratory macrophages and epithelial cells and binds to a variety of particles and bacteria [14,15]. It has been seen that the binding of potentially toxic substances, such as silica, to these receptors, causes macrophage apoptosis and the release of proinflammatory cytokines [14,16]. Comparing the hominoid genomes, two substitutions were observed for MARCO, one at position 282 (from serine in great apes, Neanderthals, and Denisova to phenylalanine in *Homo sapiens*) and one at position 452 (from histidine in primates to glutamine in Neanderthals, Denisova and *Homo sapiens*), both positively selected and altering the binding and phagocytosis functions of MARCO [15]. Therefore, some researchers have speculated that the change in aerosols breathed by human ancestors may have created a greater need for innate immunity receptors to eliminate inhaled substances, leading to genetic changes over time [2].

The discovery of fire also exposed the first humans to a new environmental stress, the smoke from burning wood. Toxic and carcinogenic substances such as polycyclic aromatic hydrocarbons (PAHs) are present in smoke. PAHs are produced by incomplete combustion or pyrolysis of organic matter and, apart from sporadic forest fires, are usually rare in the natural environment, making them a new type of environmental factor for humans [2,17,18].

Inhalation of PAHs in humans activates the aryl hydrocarbon receptor (AHR). The AHR is a transcription factor that regulates the response to xenobiotic toxicity of a number of environmental factors, such as dioxins, PHAs, and halogenated aromatic hydrocarbons (HAHs). Once bound to PAHs, AHR moves into the cell nucleus to form a heterodimer with the aryl hydrocarbon nuclear translocator (ARNT) protein. The AHR/ARNT complex binds in turn to xenobiotic response element (XRE), a specific DNA sequence located in the promoters of target genes, and activates the transcription of a number of genes, including members of cytochrome P450 family 1 (CYP1A1, CYP1A2, CYP1B1) [2,19].

Comparing the AHR of modern humans with that of Neanderthals, a valine to alanine substitution was observed in amino acid 381 of *Homo sapiens*. The ancestral AHR variant present in Neanderthals and Denisovians has a higher affinity for certain PAHs, a higher DNA-binding capacity, and 150–1000 times more CYP1A1 and CYP1B1 transcriptional activity than the human AHR. The enzymes translated by CYP1A1 and CYP1B1 mediate the metabolism of PAHs and can lead to the production of toxic intermediates. The human AHR, on the other hand, has an altered ligand binding site and thus reduced activation by PAHs, suggesting less production of toxic intermediates and greater resistance to chemicals in smoke than other hominins [2,20].

Moreover, as mentioned above, an example of genetic adaptation to toxicants is constituted by As in South America. As is a naturally occurring toxic chemical element that has been used within herbicides and insecticides in the past and is now considered a ubiquitous contaminant [21]. In certain areas of South American countries, As is of natural (e.g., mineral deposits, hot springs, rivers, and volcanic rocks) and anthropogenic (e.g., mining activities) origin. Indigenous communities in these areas have been exposed to As through drinking water for a long time. Adaptation to tolerate this toxicant has driven an increase in the frequencies of protective variants, providing evidence of human adaptation to a toxic chemical [3,22]. In fact, the way the human body excretes inorganic arsenic (iA) occurs through several stages of methylation. The process of As methylation occurs in the liver and is catalyzed by the enzyme As methyltransferase (AS3MT). Two metabolites, monomethylarsonic acid (MMA) and dimethylarsinic acid (DMA) are produced from this process and excreted in the urine along with the remaining iA residues [23]. Some studies have shown that populations that have been exposed to As for a long time have higher frequencies of the protective AS3MT haplotypes as an enhancement of As disposal from the body and adaptive mechanism [3,22,24]. For example, De Loma and collaborators [23] analyzed two indigenous communities in the Bolivian Andes (Aymara-Quechua and Uru) that are constantly subjected to the toxic agent through drinking water. This study showed positive selection for the AS3MT gene in these populations. In addition, a strong association was demonstrated between low MMA levels in urine and the frequencies of some protective AS3MT haplotypes (10 out of 35 SNPs), with a higher frequency of the protective variants for the Uru community (75%) than for the Aymara-Quechua community (66%). Among the 10 SNPs correlated with low MMA levels, the most strongly associated was found to be rs17115100 (chr10:104591393, hg19), for which carriers of the TT genotype had the lowest amount of MMA in urine (mean 7.7%). Comparing the two Bolivian communities with data on other populations, it was seen that the T allele of rs17115100 was more frequent in Aymara-Quechua (75%) and Uru (90%) populations than in other populations (e.g., Han Chinese: 35%, and Yoruba: 4%). Furthermore, the two Bolivian populations also had a high frequency (Aymara-Quechua: 73%, Uru: 82%) of the C allele of the SNP most strongly associated with low levels of MMA in urine (rs486955) compared to a previously studied population in the Argentine Andes [22]. Finally, the researchers identified a haplotype block (676 kb) in the region of AS3MT between chr10:104544753 and chr10:105221154. Within this linkage disequilibrium block, 623 polymorphisms had a strong correlation with SNP rs486955. In addition, seven of these polymorphisms were located at transcription factor binding sites, and most of them were deputed to the regulation of AS3MT expression [23].

All these data reported above are evidence of human genetic adaptation to As. Populations not adapted to As-rich environments, however, showed a higher risk of cardiovascular disease, skin cancer, bladder cancer, lung cancer, and liver cancer when exposed to this toxic agent [3].

### 2.2. Epigenetic Variability (DNA Methylation)

The phenotypic variability that can be observed among distinct individuals and populations is only partly explained by genetic variability. In fact, many studies have attempted to find direct genotype-phenotype associations to explain the visible differences between different populations, finding much less success than expected. Indeed, the transcriptome of an individual depends both on its genetics and on the regulatory events of the latter, among which are epigenetic mechanisms [25]. A prime example is monozygotic twins who, despite being genetically identical, show differences in DNA methylation and phenotype later in life, suggesting epigenetic mechanisms as regulatory intermediaries between genotype and phenotype in response to the environment [26]. Therefore, the epigenome, and particularly DNA methylation, may have been crucial in the phenotypic diversification between populations.

Recently, some studies have attempted to identify differences in DNA methylation between different populations in response to the environment and thus study the complex relationship between human phenotypic variability, DNA methylation, and environment. It was observed, for example, that West African and Northern European populations show differences in the DNA methylation of the transcription start site of genes in lymphoblastoid cell lines (LCLs) [27]. Specifically, approximately 30% of the CpG sites analyzed are differentially methylated between the two populations, and these differences correspond to more than 1/3 of the 14,495 genes studied. Furthermore, 55.4% of the differentially methylated CpG sites between the two populations are associated with a local SNP, suggesting that genetic variants are only able to explain half of the population-specific epigenetic variation. In addition, it was observed that many CpG site methylation-related SNPs (mSNPs) are able to modulate DNA methylation in a population-specific manner, possibly due to interaction with other genetic variants and/or the environment [27].

Differences in DNA methylation levels (439 CpGs) were also found between Caucasian-American, African-American, and Han Chinese-American populations, particularly in genes associated with some phenotypic differences between populations, such as response to drugs, environmental factors, and susceptibility to certain diseases (GSTT1, GSTM5, ABCB11, and SPATC1L genes). Most of the differentially methylated CpG sites were associated with genetic variation among populations, suggesting methylation as an intermediate regulatory element between genotype and phenotype, while one-third of the variations in methylation showed no association with genetic background, underscoring the independence of the epigenome and the potential of local environmental factors in shaping the epigenetic variability and thus the phenotype of populations [25].

A further study on the contribution of the environment to phenotypic differences between populations reported 916 differentially methylated CpG sites, mainly at loci related to environmental exposure, such as fetal exposure to diesel and cigarette smoke, in blood samples from four self-reported Hispanic ethnic subgroups: Puerto Ricans, Mexicans, other Latinos (all ethnicities other than Puerto Ricans and Mexicans), and mixed Latinos (individuals with the four grandparents of non-concordant ethnicities). In total, 66% of the 916 CpG sites were explained by genetic ancestry, while 34% were associated with environmental exposures, suggesting that part of the natural epigenetic variability is related to genetics and part to environmental factors, such as toxic substances. [28]. 

However, according to McKennan and colleagues, the environment does not exert such a major effect in differentiating populations through DNA methylation. In fact, the researchers identified several differently methylated CpG sites in blood samples from Black and Hispanic children but found that 66% of the CpGs associated with self-reported ethnicity were still associated with genetic ancestry. Furthermore, the researchers suggested that more than 50% of ethnicity-associated CpGs could be explained by methylation quantitative trait loci (meQTLs), which are genetic variants that influence the methylation patterns of CpG sites [29]. However, this does not preclude that some environmental factors may alter DNA methylation levels in the fetal period causing an ethnicity-related phenotype that persists with age. Indeed, analyzing some CpG sites previously identified for their response to maternal smoking in pregnancy, the researchers observed an association between DNA methylation levels and maternal cotinine levels, a substance that serves as a biomarker of cigarette smoke exposure [29,30].

It can therefore be deduced that in blood samples, population-associated methylation changes are mostly explained by the genome and that the environmental contribution to epigenetic variability could be less than expected; however, exposure to toxic substances should not be underestimated, especially during certain periods of life which are more susceptible than others [29].

The hypothesis that environmental factors, particularly toxic ones, have an impact on shaping the phenotypic variability of populations through DNA methylation is supported by a candidate gene study by Giuliani and colleagues [31]. The researchers observed an association between DNA methylation levels of blood samples from Vietnamese individuals and 2,3,7,8-tetrachlorodibenzo-paradioxin (TCDD), a component of herbicides used in the 1960s and 1970s and of Agent Orange (AO), a chemical that was used by the Americans during the Vietnam War to destroy forests in some south-east Asian countries. Specifically, the case-control study reported methylation differences in the cytochrome P450 family 1 subfamily A member 1 (CYP1A1) gene, involved in drug metabolism and xenobiotic elimination, and the insulin-like growth factor 2 (IGF2) gene, involved in development and growth. In addition, individuals whose parents resided in the former spraying areas showed methylation differences in the CYP1A1 and IGF2 genes, suggesting a response to TCDD exposure in offspring of parents that were exposed during the Vietnam War.

Toxic substances are thus capable of modeling DNA methylation profiles and leading a part of the population to develop a phenotype better suited to the historical event/environment undergone [31].

Among the various toxic substances to which humans are exposed, there is As. While some populations are genetically adapted to As, as described above, others show variability in terms of DNA methylation levels. A study by Alegría-Torres et al. measured the DNA methylation levels of Mexican children aged from 6 to 12 years residing in areas where mines had long been present. Specifically, methylation levels of Alu and LINE-1 in the blood were found to be associated with As concentrations in the children’s urine. In particular, LINE-1s tended to be hypomethylated [32].

A further study conducted by González-Cortés and colleagues on Mexican children exposed to As in fetal or infant age showed hypermethylation of the matrix metalloproteinase-9 (MMP9) gene promoter in children most exposed to As. In addition, a positive correlation was found between As levels in children’s toenails and DNA methylation levels of MMP9. Furthermore, positive associations were found between both receptor for advanced glycation end products (RAGE) gene methylation levels and iAs and DMA concentrations in children’s urine, and tissue inhibitor of matrix metalloproteinase—1 (TIMP1) gene methylation levels and first As methylation. Finally, the researchers suggested that alterations in methylation levels of extracellular matrix remodeling genes caused by As could, in turn, lead to the onset of lung disease [33] and increased risk of cancer. Indeed, it has been observed that one of the underlying causes of As carcinogenicity is aberrant DNA methylation [34]. One of the first studies to support this theory was led by Zhao and coworkers, who found a positive association between cancer occurrence and loss of global DNA methylation following As exposure of rat liver cells [35]. Among the molecular mechanisms at the center of this carcinogenic process could be the deficiency of SAM, which is used for As biomethylation and consequently is insufficient for normal DNA methylation processes [34]. In addition, As causes DNA damage through increase in reactive oxygen species (ROS), and this, in turn, interferes with the activities of DNMTs. Finally, the further synthesis of reduced glutathione within cells, caused by increased ROS levels, leads even more to the blockage of SAM synthesis.

However, some studies attribute the loss of methylation to a direct action of As on DNMT proteins since they are rich in sulphydryl, a free radical that maintains the stability of the protein structure, and are therefore subject to major alterations by this toxicant [34]. Indeed, it has been shown that exposure to As, even in the case of AS3MT deficiency, induces a reduction in DNMT enzyme activity [35,36] and in the mRNA expression of DNMT1 and DNMT3A [37,38].

Besides the loss of methylation, As also causes loss of expression of some genes, such as p53, RASSF1A, DAPK, SLC7A7, PPBP, MRPL24, MRPS5, CREBBP, ADAP2 [21,38,39], and hypermethylation of CpG sites in SEC31A, LOC144776, KRTAP2-2, TFR2, RPL9P23, and MST1P2 genes, implicated in cell death and cancer [39].

Lastly, cord blood samples from children of Mexican origin exposed to As through drinking water also showed 2919 differentially methylated genes. Among these genes, 16 were enriched for transcription factor binding sites (EGR and CTCF), and 7 had methylation levels correlated with differences in gestational age and head circumference at birth [40].

## 3. Toxicants Exposure, DNA Methylation, and Mutation Rate

Many studies question the real impact of DNA methylation on human evolution, claiming as evidence the lack of consensus on transgenerational inheritance studies of the mechanism. It is not the purpose of this review to enter into this dispute for which there are already numerous papers [41], but we believe it is important to describe the mechanisms by which DNA methylation influences the mutation rate. Below, we thus reported evidence that showed that DNA methylation dynamics has a mutation impact from methylated CpG to TpG, and this also occurs in the germline [42,43]; therefore, DNA methylation constitutes a mechanism of uncontroversial importance for human evolution.

In the last decade, many studies showed that endocrine disruptors do not promote genetic mutations directly but indirectly via epigenetic alterations [44], suggesting that transgenerational alterations in the epigenome increase genetic instability and promote genetic mutation and variation [45].

Several studies have identified methylation of CpG sites as predictive of higher mutation rates. This occurs because, normally, spontaneous deamination converts unmethylated cytosines to uracil (C > U) and, subsequently, the enzyme uracil-DNA glycosylase removes uracil from the DNA sequence. In the case of methylated cytosines (mC), however, the process of spontaneous deamination converts cytosines to thymines (mC > T), with a rate of 2–3 times higher than unmethylated cytosines [46]. Thymine cannot be eliminated from the sequence, such as uracil, thus making methylation of CpG sites a major reason for the increased rate of C > T mutations at those sites [47,48,49] (Figure 2). Furthermore, the prevalence of methylation in CpG dinucleotides results in a large loss of these sites due to the high mutation rate that follows. In this way, DNA methylation has a direct effect on genome mutation and, thus, on human evolution [49].

Recently, some studies have shown how the rate of point mutations in germ cells can be affected by the expression level, replication times, the methylation level of CpG sites, and the number of CGs [43].

For example, Agarwal and collaborators [48] focused on 1.1 million methylated CpG sites present in the exons of the ovaries and testes, considering a group of about 2900 parent-offspring trios. The average C > T mutation rate of these sites per generation was calculated, and subsequently, the authors evaluated a larger sample of 350,000 CpGs carrying synonymous mutations, which are evolutionarily neutral as they replace the final amino acid with an amino acid with similar properties. The study showed that almost all (~99%) of the methylated CpG sites segregated a T in the absence of selection, so not observing a T is a clear indication of strong purifying selection. Methylation of CpGs would therefore increase the probability of mutation saturation at these sites and could be useful in identifying CpG sites with evolutionarily non-neutral mutations. Furthermore, the strength of natural selection on some mutations could be related to their pathogenicity [48].

Another recent study by Zhou and colleagues [43] investigated the association between dynamic levels of DNA methylation during germinal development and germline mutation rates. The authors used whole-genome methylation data from 13 stages of germinal development and rare mutation data (51,739,442 SNPs) from the 1000 Genome Project. The results of this study confirmed that methylated CpG sites tend to be more likely to mutate than unmethylated CpG sites. A clear example of this can be seen by observing the sperm stage, during which the methylated CpG sites had a mutation rate of 17.73%, while unmethylated CpG sites had a mutation rate of 4.92%. Furthermore, the researchers reported that DNA methylation levels, both at the level of individual CpG sites and at the regional level, were positively correlated with the rate of CpG C—T point mutations and with the mutations rate of genomic (more so in intronic regions) and chromosomal regions (mainly chr. 9, 11, 14, and 17), respectively. In addition, some gene expression data from embryonic tissues were analyzed, considering that DNA methylation is negatively associated with gene expression. This analysis showed that the association between DNA methylation levels and mutation rate at the gene promoter level was more significant in germ tissues than in somatic tissues, suggesting that methylation at CpG sites plays a key role in modulating the rate of DNA mutation in the germ line, and thus disease occurrence and human evolutionary adaptation [43].

Considering that DNA methylation may be subjected to environmental factors, it is likely that exposure to certain substances, such as chemicals, endocrine disruptors, or toxic agents, may be the cause of increased mutation rates (and thus of certain diseases) within a population through alteration of DNA methylation.

An example of this phenomenon is provided by Zhang and colleagues [50] in a study focused on arsenism, a condition caused by chronic exposure to As. DNA methylation levels were measured in the p53 gene promoter and exon 5 of the p53 gene, while the mutation rate was checked in exon 5 of the p53 gene. The results showed a positive rate of methylation at the promoter level of the p53 gene of 13.16% (individuals with mild arsenism), 27.91% (individuals with moderate arsenism), and 45.16% (individuals with severe arsenism), compared with the control group (1.11%). Regarding the methylation rate in exon 5 of the gene, individuals with low exposure reported a rate of 55.26%, those with medium exposure 51.16%, and those with high exposure 48.39%, while controls reported a methylation rate of 88.88%. The mutation rates of exon 5 of the p53 gene were found to be 5.26% (mild exposure), 16.28% (medium exposure), 25.81% (high exposure), and 0% (controls). In addition, an association was observed between the mutation rate in exon 5 of the p53 gene and both gene promoter hypermethylation and exon 5 hypomethylation. The researchers concluded that toxic substances, such as As, are able to interfere with DNA methylation by increasing the mutation rate in individuals exposed to this type of environmental stress [50].

A recent study [51] focused on fetal exposure through the maternal placenta to air pollution (PM_2.5_, black carbon, and NO_2_), specifically analyzing its impact on DNA methylation levels and mutation rate in the placenta. The level of DNA methylation in the promoter region of seven DNA repair and anti-tumor genes was measured in a cohort consisting of mother-newborn pairs. In addition, the mutation rate of Alu, a mobile element that serves as a biomarker for general DNA mutation levels, was assessed. This study showed a positive correlation between a high Alu mutation rate and increased exposure to fine particulate matter and black carbon. DNA methylation analysis showed a positive correlation between increased PM_2.5_ exposure and promoter hypermethylation of most of the considered genes. For NO_2_ exposure, however, there was no increase in the rate of Alu mutation or methylation in gene promoters. The researchers concluded that exposure at fetal age to some components of air pollution can alter the methylation level of some genes important for DNA repair, thus causing an imbalance in the body’s defense mechanisms and increasing the overall DNA mutation rate (Alu). Exposure to these types of pollutants results in the onset of problems in DNA damage repair in the fetus/infant and an increased risk of cancer occurrence later in life [51].

## 4. Link between Toxicants Exposure, DNA Methylation, and Biological Fitness

Recently, the increased human exposure to chemicals—due to growing industrialization—has been related to reproductive disorders showing its potential to influence fitness and, thus, human evolution itself.

Biological fitness, from an evolutionary point of view, indicates the ability to survive and transmit genes to the next generation (also known as “reproductive fitness”) [52,53].

In general, biological fitness can be enhanced in two ways: by increased fertility of the individual, i.e., increasing the number of offspring, and/or by increasing the number of surviving offspring (reproductive success) [54]. Evolutionary studies showed the complexity of this process that involves both physical, cultural, and behavioral components; for example, the presence of excessively masculine traits, likely due to an elevated testosterone level, would tend to suppress the male propensity to care for offspring, leading to a decrease in reproductive fitness in individuals with such traits [54,55], suggesting complex biological and behavioral interactions.

Exposure to various chemicals, such as endocrine disruptors, heavy metals, radiation, or cigarette smoke, cause impairment in terms of biological fitness as they directly impair fertility, in terms of poor sperm quality, miscarriage in the early stages of gestation, and loss of the pre-implantation embryo [56].

Given the property of DNA methylation mechanisms to respond to environmental stresses, a correlation between exposure to polluting chemicals and alterations in DNA methylation may exist, with consequent effects on biological fitness. Moreover, DNA methylation signature profiles are also associated with the selective survival of embryos in hostile prenatal environments, as demonstrated in the Dutch famine study [57].

The nature of contaminant substances is very broad, and not all of them have been thoroughly studied. Recently studies on Endocrine Disrupting Chemicals (EDCs), a collection of natural or synthetic substances ubiquitous in the environment, are increasing, as EDCs interfere with normal human endocrine function and alter DNA methylation, with a consequent strong impact on reproductive fitness [8,58,59,60].

EDCs at low levels do not constitute a risk for humans, but if their concentration increases, their impact on human biology becomes not negligible [61]. Furthermore, exposure to these chemicals can occur in adulthood but also in childhood or during gestation, leading to consequences later in life [8].

### 4.1. Data in Males

The increased presence of chemicals in the environment and their potential role in interfering with DNA methylation and human reproductive health are part of a broader scenario in which an increase in world infertility is observed. With regard to males, over the past 50 years, there has been a reduction in the number and in quality of sperm [62]. According to data from the World Health Organization (WHO), in 2008, the rate of male infertility (poor sperm mobility, reduction in sperm count, damage to sperm DNA) among infertile couples was around 30% worldwide, making male infertility one of the most common reproductive disorders [62,63,64].

The development of spermatozoa involves a series of molecular and morphological modifications that primordial germ cells (PGCs) have to face and where DNA methylation plays a relevant role.

Sperm quality, and the subsequent embryonic development, depend on various sperm epigenetic modifications/alterations that can occur during spermatogenesis [65]. A study by Marques and collaborators reported hypo and hyper-methylation of the imprinted genes H19 and MEST in sperm from oligospermic individuals. Furthermore, non-methylation of the CTCF-binding site was also observed, which could lead to poor embryo quality [66]. These results were confirmed by another study, which reported methylation errors in H19-DMR and PEG1/MEST-DMR of oligospermic men [67]. In addition, a further study observed an association between low sperm quality and hypermethylation of different sperm DNA sequences, suggesting errors in the epigenetic reprogramming step during spermatogenesis [68].

These kinds of epigenetic alterations, such as aberrant DNA methylation, have been associated with the surrounding environment as spermatogenesis is characterized by a strong reprogramming of the epigenome and, therefore, more subject to external disturbances [65,69,70].

Several studies have shown that there are different toxic substances that can alter sperm DNA methylation, influencing reproductive fitness and contributing to the increase in male infertility.

The organochlorine pesticide para-dichlorodiphenyltrichloroethane (DDT) is categorized among EDCs and is an example of how exposure to toxic agents, DNA methylation, and fitness are closely related. DDT has been banned in many countries but is still used to combat malaria in some developing countries [71]. Evidence of its effect on reproductive fitness is provided by a study by Skinner and colleagues [72], in which pregnant F0 female rats were exposed to DDT, and then the next-generation diseases were analyzed. Males from the F1 generation, exposed to DDT in utero, showed an increase in prostate disease in adulthood, and males from the F3 generation, indirectly exposed to DDT, reported an increase in testicular disease (azoospermia, vacuoles, and exfoliated germ cells in the seminiferous tubules, lack of lumen in the seminiferous tubules), an increase in sperm cell apoptosis and fewer spermatozoa. According to this study, testis diseases were transmitted through the female germ line. Furthermore, in the F3 generation sperm, differently methylated regions (DMRs) unique to the DDT exposure were found [72]. All the conditions observed cause difficulties in generating an adequate number of offspring and therefore cause a lower reproductive fitness.

Particularly dangerous is the DDT metabolite p,p′-dichlorodiphenoxydichloroethylene (p,p′-DDE). As studied by Song and colleagues, rat exposure to p,p′-DDE during pregnancy can lead to hypomethylation of IGF2 in sperm, causing an alteration of testicular histology and male fertility [73]. In addition, a recent human study showed that two other metabolites of DDT (2,2-bis(p-chlorophenyl)-acetic acid (DDA) and 1-chloro-2,2-bis-(p-chlorophenyl) ethylene (DDMU)) can interfere with the DNMT1 activity by binding to it and modifying its catalytic domain conformation, resulting in genomic hypomethylation and alteration of the methylation level of the sexual development genes promoters and of the expression of the Sox9 and Oct4 genes in the embryo, with probable negative influence on offspring survival and health [74].

The fungicide Carbendazim (CBZ) also acts as an endocrine disruptor [75]. This chemical is widely used in agriculture on a wide range of crops (such as cereals, vegetables, cotton, and tobacco), as it limits the presence of ascomycetes and basidiomycetes. Low doses of CBZ can lead to the interruption of spermatogenesis, causing a reduction in the number and motility of spermatozoa, and to the reduction of DNA methylation level in Leydig cells, interstitial cells of the testes that constitute the primary source of testosterone in males [76].

Vinclozolin (VCZ) is also categorized among EDCs. Vinclozolin is a fungicide used in agriculture to protect crops from diseases caused by fungi. Since 2006, this fungicide has been banned in several countries as it proved to be an endocrine disruptor with adverse potential for human health. Vinclozolin fungicide causes alterations in F0 pregnant female rats during the gonadal sex-fixation phase, and subsequent generations (F1–F4) showed prostate diseases (prostatic lesions and prostatitis) and testes alterations (increased apoptosis of spermatogenic cells, abnormal spermatogenesis, and subfertility). These anomalies appear to be correlated with epigenetic modifications of male germ cells, such as aberrant DNA methylation [77]. A reduction in methylation levels of the H19 and Gtl2 genes and an increase in the DNA methylation level of the Peg1, Snrpn, and Peg3 genes for the F0 male generations exposed to VCZ in utero has been described [78]. The sperm concentration of the offspring was assessed, and a decrease in sperm count was found for the F1 generation treated with VCZ, underlining a negative effect of this substance on male reproductive health, possibly through the inference of errors in imprinting during spermatogenesis [78].

Bisphenol A (BPA) is another estrogenic endocrine disruptor used to make polycarbonate plastics and epoxy resins, which can contaminate food and water. It has been demonstrated that neonatal exposure of rats to BPA can interfere with reproductive function by altering DNA methylation. In particular, early exposure to BPA is associated with significant hypermethylation of Estrogen Receptors alpha (ERα) and beta (ERβ) promoter regions, which is also maintained in adult testis, indicating a negative effect of this chemical on spermatogenesis, resulting in impaired fertility in adulthood [79,80,81]. Furthermore, it was observed that exposure to BPA during gestation and lactation can cause a decrease in fertility through DNA hypermethylation of the DNMT3A promoter, whose expression is linked to global DNA methylation [80]. Indeed, exposure of the mouse GC-1 spermatogonial cell line to a low dose (1 µg/mL) of BPA leads to a slight decrease (21.2%) in global DNA methylation, whereas exposure to a high dose of BPA (10 µg/mL) corresponds to a significant twofold reduction in global DNA methylation levels, and to a decrease in DNMT1 protein and mRNA expression levels, resulting in global DNA hypomethylation and alteration of normal spermatogonial development and offspring survival [82]. These alterations also occur at the candidate gene level; in fact, modifications have been identified in the DNA methylation level of the myosin-binding protein H (mybph) and of the protein kinase C δ (prkcd) following exposure of mice to BPA. This has led to a reduction in the proliferation and motility of spermatocytes and, therefore, to fewer offspring [83]. Findings on mouse models are also supported by human studies. Indeed, data about a group of male factory workers exposed to BPA showed the toxic effect of this chemical compound on the methylation status of Long Interspersed Nucleotide Element-1 (LINE-1) in spermatozoa. The results of the case-control study indicated lower LINE-1 methylation level and lower sperm quality in exposed workers, resulting in poor reproductive health [84].

Among the endocrine disruptors, there are also polychlorinated biphenyls (PCBs), which are chlorinated hydrocarbons used for the most part at industrial level, as they have non-flammable conductive, and insulating properties. These compounds have high toxicity and carcinogenicity, and for this reason, they have been less and less employed since the 1960s. A recent study evaluated the effect of PCBs on gestating mice. 2,3′,4,4′,5-pentachlorobiphenyl (PCB118) showed effects on the reproductive health of male fetuses exposed directly to this chemical through the maternal placenta. The study demonstrated a global DNA hypomethylation and a lower expression of DNMT1, PCNA (proliferating cell nuclear antigen), a protein that interacts with DNMT1 to help and maintain the methylation process, and stimulated by retinoic acid 8 (STRA8), a PCNA-dependent spermatogenesis regulatory factor, in the testes of the offspring exposed to the toxic agent. The researchers suggested that the decrease in DNMT1 and PCNA expression, and the consequent decrease in their interaction, may have led to the reduction of overall DNA methylation in the testes of mouse embryos, with possible consequences on their health and fertility. The analyses on 7-week-old mice showed that the alterations of DNA methylation, which occurred during the embryonic stage, were also maintained after sexual maturity, highlighting the possibility of a lower reproductive capacity in adulthood. Indeed, this condition coexists with low sperm quality (in terms of sperm motility and deformation rate) and an increase in the diameter of the seminiferous tubules, underlining how prenatal exposure to PCB118 can lead to infertility through epigenetic modification [85].

Another cause of male infertility is exposure to excessive heat. A mouse study showed that prolonged exposure of the scrotum to heat stress (39 °C) can cause alterations in the spermatic methylome (DMRs). In addition, it was found that the Pik3cg and Nr4a1 genes, which are hypermethylated in paternal sperm, are also hypermethylated in the offspring’s liver, with consequences for the protein expression of these genes. Certain environmental factors acting on DNA methylation can, therefore, both decrease male fertility and cause health problems in the next generation [86].

The above-mentioned studies refer to the effects of single pollutants on male reproductive fitness, but there are many studies that reported the detrimental impact on the fitness of mixtures of compounds, as in the case of Persistent Organic Pollutants (POPs).

POPs are organic compounds that contain a part of artificial chemicals, which accumulate in the environment and pose a risk to both the environment and human health. In a recent study, Maurice and colleagues hypothesize that direct exposure to POPs can alter DNA methylation levels, leading to a series of consequences on reproductive fitness in adulthood. This work indeed highlights an association between the exposure of rats to POPs in utero and during lactation and the decrease in male fertility through aberrant DNA methylation. In particular, the male F1 generation, directly exposed to these chemicals through the uterus or breast milk, showed a reduction in testosterone levels, low sperm quality, production, and motility. These are all traits that could lead to a lower probability of mating and a reduced number of puppies. Furthermore, exon 10 of the DNMT3L gene was hypermethylated in rats early exposed to POPs. This modification leads to the Dnmt3l inactivation, which in turn causes modifications in DNA-methyltransferases 3 alpha and beta activity, with consequent defects in spermatogenesis. In addition, the hypermethylation of the T-box transcription factor 2 (Tbx2) gene and the hypomethylation of the Receptor associated protein of the synapse (Rapsn) gene were observed in sperm DNA. In general, there was also a reduction in the number of puppies per pregnancy and an increased loss of the embryo before implantation in endometrium for the F1 generation exposed to POPs [87]. Another rat study supported the above-mentioned hypothesis showing that prenatal exposure to POPs can impair male fitness, altering embryo gene expression and damaging sperm quality [88]. Specifically, F1 generation germ cells were directly exposed to POPs through the uterus of exposed F0 mothers. The F1 spermatozoa showed a reduction in their vitality and motility, with possible consequences on the quantity and health of the offspring.

A recent human study evaluated the effect of a mixture of chemicals on the inhabitants of the Faroe Islands (Denmark). Fish and foods contaminated with various chemicals are widely consumed on these islands. Researchers examined genome-wide DNA methylation of cord blood samples and verified the presence of an association between DNA methylation levels and a number of toxic substances, specifically Methylmercury (MeHg), major PCBs, Hexachlorobenzene (HCB), p,p′-DDE, p,p′-dichlorodiphenyltrichloroethane, and Perfluoroalkyl substances. It was shown that most of the differentially methylated CpG sites were associated with PCB105. In addition, changes in DNA methylation were different between men and women. In addition, some of the differentially methylated male sites were enriched in X-chromosome cytobands, suggesting sex-specific responses of the epigenome to early chemical exposure and different consequences on the health of the offspring exposed to the mixture of contaminants [89].

Air pollution is a mixture of different pollutants of various origins, such as coarse particulate matter (PM_10_), fine particulate matter (PM_2.5_), and some gases. The data that link air pollution to DNA methylation and reproductive outcome are emerging. For example, Yauk and colleagues investigated the effect of air pollution on gametes of male mice, reporting an increase in sperm mutation frequency, DNA damage, and methylation in mice exposed to air pollution, with possible consequences on the reproduction and health of their offspring [90]. The effect of air pollution on biological fitness and its relationship to DNA methylation were also explored by another study on a mouse model, from which emerged that long-term (40 days) exposure of adult mice to fire smoke causes changes in sperm DNA methylation, with potential reproductive risks [91]. These findings have also been confirmed by human studies. For example, a recent study on a large cohort of men investigated the effect of long-term (1 year) exposure to air pollution, with particular attention to the impact on reproductive fitness in terms of sperm quality. It emerged that long exposure to single pollutants (PM_10_, PM_2.5_, SO_2,_ and NO_2_) was negatively associated with sperm motility. In addition, it was found that co-exposure to six air pollutants was related to a decrease in sperm motility and, therefore, to a poor chance of large offspring. In particular, exposure to PM_10_ led to a decrease in sperm DNA methylation [92].

For clarity, we summarize what is described in this section in the following table (Table 1).

### 4.2. Data in Females

Although male infertility is widespread, female infertility appears to be more prevalent [64,93]. In contrast to males, in the female sex, the oocytes remain demethylated until the arrival of puberty. This aspect, therefore, also makes the childhood period a critical time frame because of the vulnerability of the epigenome to environmental exposure, possibly affecting female reproductive health.

Regarding females, the available data linking DNA methylation, exposure to harmful chemicals, and reproductive fitness are limited and reported below.

Methoxychlor (MXC) is a pesticide used in agriculture, and it is categorized as EDC, as it exhibits estrogenic, antiestrogenic, and antiandrogenic properties. In 2009, fetal and neonatal rats were exposed to MXC, and subsequently, the methylation level of ERα and ERβ gene promoters in the ovaries was measured [94]. This analysis revealed an association between DNA hypermethylation of the ERβ gene promoter and exposure to MXC. Within the same study, the level of global DNA methylation in the ovaries was also measured, showing hypermethylation in specific genes in rats exposed to the MXC. Finally, the researchers found an increase in the Dnmt3b expression level in the ovaries of the exposed rats, related to the increase in DNA methylation level reported above. This work therefore suggested that exposure to MXC in the early stages of life can cause ovarian dysfunction in adulthood through modification of methylation profiles and, thus, alterations related to fertility [94,95]. A case-control study subsequently conducted by the same authors investigated the effects of administering different doses (20 μg/kg/day or 100 mg/kg/day) of MXC on the ovaries of female rats. Specifically, MXC was administered to pregnant female rats from day 19 to day 22 of embryonic development and to offspring from day 0 to day 7 of postnatal life. Subsequently, the DNA methylation level in the ovaries of female rats was measured at day 7 and day 60 of life (adult specimens). Genome-wide methylation analysis in ovaries exposed to the chemical agent registered increased DNA methylation at some loci encoding molecules typical of PTEN, IGF-1, and Rapid estrogen signaling pathways, which was also associated with ovarian defects and infertility in female rats [96].

Other substances, such as Endosulfan, may influence reproductive fitness by causing DNA methylation alterations. Endosulfan is a highly toxic organochlorine insecticide and acaricide, classified as an EDC due to its estrogenic activity. This chemical is banned globally, but it is still used illegally in some developing countries [97,98]. Recent studies on female rats have focused on the effects of Endosulfan on the ERα signaling pathway and the Homeobox A10 (Hoxa10) gene linked to the uterine receptivity of the embryo. The studies reported DNA methylation defects in the regulatory regions of the Hoxa10 gene, decreased expression of Hoxa10, and increased expression of Dnmt3a and Dnmt3b in utero in rats treated with Endosulfan. In addition, these rats showed increased expression of ERα and DNA hypomethylation in their regulatory regions. As a result, the expression of ERα-dependent genes (MUC-1 and IGF-1) appeared to increase due to the ERα overexpression. These results suggest that errors in DNA methylation of the Hoxa10 gene and aberrant expression of ERα caused by exposure to Endosulfan could be the cause of embryonic losses prior to implantation in the uterine endometrium and thus of infertility or fewer puppies [98,99].

Moreover, exposure to PCB118 may impact reproductive fitness also in females influencing the possibility of preimplantation embryo loss. Specifically, female mice were exposed to PCB118 and, after mating, were killed on day 4.5 of pregnancy to study the effects of the chemical agent on female reproductive health. The case-control study reported increased abnormal uterine morphologies and embryo implantation failures in the uterine endometrium in PCB118-treated females. In addition, downregulation of Hoxa10, integrin subunit beta, and estrogen receptor 1 (ER1) genes were reported. It is likely that the downregulation of the Hoxa10 gene promoter, which probably increases preimplantation loss and infertility in females, is due to an increase in DNA methylation level [100].

Opposite outcomes were observed during the study of the negative effects of BPA (mentioned above) on female fertility. In fact, a study on the murine model showed that prenatal exposure to BPA can lead to a reduction in methylation level of the promoter and intron of the Hoxa10 gene, which regulates the development of the embryo’s uterus, interfering with female reproductive capacity, and is necessary for the implantation of the embryo into the adult endometrium [101].

Early exposure to the previously cited DDT also has effects on female fitness. Indeed, it has been shown that exposure to DDT of later rat generations through the maternal uterus (F0 generation) results in an increased presence of polycystic ovary in F1 generation females, which is a condition that can interfere with normal ovulation and, therefore, with fertility. In addition, despite not being directly exposed to the toxic agent, F3 generation females show increased uterine infection (uterus enlargement, presence of purulent material, and inflammation) [72].

The effect of DDE also has an impact on female reproductive capacity. A study by Cohn et al. focused on the analysis of p,p′-DDT and p,p′-DDE concentrations in maternal blood serum samples taken a few days after partum from women who gave birth in the United States in the early 1960s. This study revealed an association between high concentrations of the pesticide DDT in the serum of the women and the shorter pregnancy time of daughters of the same women, about 30 years later. In particular, the chance of getting pregnant in the daughters decreased by 32% at 10 microg/L of p,p′-DDT in maternal blood [102].

Finally, another factor affecting female fertility is the accumulation of oxidative stress resulting from the growing trend to delay childbearing, observed especially in developed countries with longer life expectancy. This example is not directly related to exposure to external toxicants analyzed in this review, but it is indirectly connected as, in this case, the “toxicants” are of endogenous origin and related to the process of aging (for details on how aging may increase spillover of self-garbage molecules see [103]). Specifically, the oocytes of mature women are exposed to oxidative stress for a longer period of time than those of younger women, with negative consequences on the development and quality of the oocytes and, thus, on fertility [104]. In particular, a recent study on mice reported differences in methylomes of young and old mice during oocyte maturation and an upregulation of the expression level of Dnmt3a and Dnmt3l in immature oocytes of old mice [105].

For clarity, we summarize what is described in this section in the following table (Table 2).

## 5. Link between Toxicants Exposure, DNA Methylation, and Survival

As mentioned above, fitness is not only related to the ability to procreate and pass on genes to the next generation, but it also refers to the ability to survive.

In this section, we review the recent literature regarding the impact of toxic substances or chemicals on fitness, in terms of survival and mortality, through DNA methylation, a valid and robust biomarker of accelerated biological aging. In fact, one of the most effective and widely used tools for evaluating the biological age of people is represented by “epigenetic clocks”, which are the result of the combination (through validated mathematical algorithms) of DNA methylation levels at specific CpG sites that change in a predictable way over the lifespan. Deviations between epigenetic age and chronological age can predict a broad range of pathological conditions and mortality risk [106]. The first epigenetic clock was developed by Horvath in 2013 [107], and many more have been developed since then [108,109,110,111].

It is worth mentioning that epigenetic acceleration may not always be related to fitness because much depends on timing and tissue. Epigenetic age acceleration that occurs only late in life (after reproductive age) is likely negligible in terms of fitness, while epigenetic acceleration observed early in life may play a major role. Moreover, epigenetic aging that has been found in female reproductive organs may negatively contribute to reproductive outcomes. Mother-accelerated epigenetic clocks were associated with shorter gestational length and lower birth weight, two leading causes of neonatal death [112].

A recent study evaluated the effects of industrial solvents benzene and trichloroethylene on Chinese workers exposed to these two carcinogens. Specifically, leukocyte DNA methylation was measured from blood samples, and then, five epigenetic clocks were calculated, together with DNA methylation-based telomere length (DNAmTL). Workers exposed to the two carcinogens showed an increased epigenetic age acceleration, according to the Skin-Blood Clock [113].

Another recent study, on the other hand, evaluated the impact on human health and aging of exposure to various conditions/substances during early fetal life and childhood. Briefly, the researchers investigated the link between a large number of exposomes, including cigarette smoke, toxic chemicals, and air pollution, and epigenetic age acceleration, calculated with the Skin and Blood clock from a large number of blood samples belonging to children in infancy (7 years old). The results of this study reported an association between accelerated epigenetic aging and in utero exposure to cigarette smoke during the fetal phase of the child. Increased age acceleration was also reported in children exposed in infancy to parental smoking and indoor air pollution. In contrast, exposure to certain chemicals, such as polychlorinated biphenyl-138 (PCB-138) and the pesticide dimethyl dithiophosphate (DMDTP), has been shown to be associated with a deceleration of epigenetic aging. However, it is worth noting that only the “epigenetic aging-exposure to indoor air pollution” and “epigenetic aging-exposure to DMDTP” models were significant (*p* < 0.05) after a series of adjustments with respect to maternal body mass index (BMI) before pregnancy, birth weight, and infant BMI [114].

Within this type of studies, some evidence has reported air pollution as impacting epigenetic age. A study by White and collaborators, for example, evaluated the effect of nitrogen dioxide (NO_2_) and particulate matter (PM_2.5_ and PM_10_) on accelerating epigenetic age. For this purpose, blood samples were taken from women exposed to air pollution, and then DNA methylation was measured, and epigenetic age was calculated using three epigenetic clocks. Regarding NO_2_, the results of the association study reported a negative association with accelerated aging. The association of PM_2.5_ with epigenetic age acceleration, on the other hand, varied according to the type of components. In fact, an acceleration of epigenetic age of about 6 years for PM_2.5_ with a high presence of crustal elements, an acceleration of about 2 years for PM_2.5_ with low sulfur, and a deceleration of about 1 year for PM_2.5_ with few nitrates were observed. Finally, no associations with epigenetic age were found for PM_10_. Overall, the researchers concluded that some components of air pollution are able to interfere with the epigenetic age, thus influencing mortality risk [115].

These findings are in line with those of a previous study on fine particulate matter, which reported an association between the increase in certain components (sulfate and ammonium) of PM_2.5_ and the epigenetic age acceleration of about 0.4 years calculated through Horvath’s epigenetic clock [116].

Lastly, a recent study on smoking, using five epigenetic clocks, showed an association between epigenetic age acceleration, calculated from genome-wide DNA methylation levels of blood samples (leukocytes), and cigarette consumption. In particular, some clocks reported, for former smokers, an epigenetic age acceleration ranging from 0.77 to 2.69 years (depending on the considered clock) compared with non-smokers. In addition, to take into account possible changes in blood cell composition that may affect epigenetic age estimation, extrinsic epigenetic age acceleration (EEAA) was calculated. The EEAA reported an increase of 1.27 years in smokers compared with never-smokers. Then, intrinsic epigenetic age acceleration (IEAA), which is based on Horvath’s clock and whose measurement is related to cellular aging, was also calculated. IEAA reported an increase of 1.03 years in smokers compared with no smokers. Furthermore, former smokers had shorter telomere lengths than never-smokers. For current smokers, on the other hand, differences in DNA methylation in 46 CpGs sites and 5 regions were observed when compared with non-smokers. These results could be interpreted as indicating that exposure to harmful substances, such as cigarette smoke, may lead to an acceleration of the aging process [117].

This study is supported by another one recently conducted by Bozack and co-workers on exposure to As, a toxic chemical element that is ubiquitous due to anthropogenic and natural causes. The acceleration of epigenetic age was calculated from the methylation levels of buccal and peripheral blood cells using different clocks. Telomere length is also estimated from the DNAm data.

Regarding blood cells, early exposure to As resulted in an acceleration of epigenetic age, and the correlation is also maintained after adjusting for cell type. On the other hand, for buccal cells, no statistically significant correlation was reported [118].

The totality of these data suggests that, in general, exposure to toxic substances may affect the epigenetic aging trajectory from the first stage of life and during the life course, but there are also studies that reported contradictory findings.

An example is an effect of lead (Pb) exposure on epigenetic aging. Pb is a heavy metal pollutant, hazardous to human health, and ubiquitous in the environment, as it is present in drinking water, air, food, cigarettes, fuel, paints, pesticides, and many other tools or processes of industrial or domestic origin.

Specifically, the researchers investigated the relationship between intrauterine fetal Pb exposure, telomere length, and epigenetic aging trends (Horvath’s DNA methylation age and Knight’s predictor of gestational age) through analysis of cord blood (leukocyte) samples and postnatal bone Pb levels. No significant associations were found between these data and Pb levels in children’s bones, pointing out no consequences at the level of epigenetic aging for infants exposed intrauterinally to Pb [119].

In addition, Xu and colleagues have focused on the effect of perfluoroalkyl substances (PFAS), which are a group of POPs that are highly toxic and persistent in the environment and the human body. They investigated the relationship between the exposure of Swedish women to PFAS-contaminated water and the acceleration of their epigenetic age calculated by Horvath’s epigenetic skin and blood clock. Women were divided into groups based on the degree of exposure to the toxic substances. DNA methylation differences were reported among the different groups of PFAS-exposed women, related to genes associated with cancer, endocrine and reproductive diseases, and other signaling pathways. However, even in this study, no significant differences in epigenetic age acceleration were observed, suggesting that PFAS are capable of causing aberrant methylation patterns but do not act on the epigenetic aging acceleration [120]. Nevertheless, it must be considered that different epigenetic clocks have been developed, and sometimes different clocks can return different results.

## 6. Link between Toxicants Exposure, DNA Methylation, and Brain Evolution

The evolution of the brain has definitely characterized the natural history of humans. In *Homo sapiens*, the size of the adult brain is the result of a process of encephalization that has profoundly influenced the entire biology of the species and, above all, the physiological processes associated with its growth and development, both in prenatal and postnatal life. DNA methylation plays a critical role in human brain development, regulates synaptic plasticity, learning, and memory functions, and is implicated in the pathogenesis of several neurological diseases [121,122]. Not surprisingly, DNA methylation has also played a key role in brain evolution, as recently confirmed by a recent study [123]. By comparing brain DNA methylation in humans, chimpanzees, and rhesus macaques, Jeong and co-workers identified human-specific epigenetic patterns that occurred during evolution in a cell type-specific manner. Genomic regions with human-specific DNA methylation tend to be hypomethylated compared to a chimpanzee, highlighting a greater transcriptional and expressive capacity of human brain DNA. Moreover, these genomic regions are deeply implicated in brain regulatory functions and are associated with an increased genetic risk of developing neuropsychiatric disorders [123].

Given the potential of DNA methylation in shaping brain development and promoting the onset of neurological disorders, it is plausible to speculate that today’s human exposure to industrially produced chemicals, which are adept at altering DNA methylation profiles, may be the cause or concomitant cause of the development of neurological/neurodegenerative diseases.

Accordingly, studies analyzing the link between toxicants exposure, DNA methylation, and the onset of neurological diseases have been recently published [124,125,126].

Parkinson’s disease (PD) is among the most studied neurodegenerative diseases. This pathology mainly causes loss of control in movement and loss of balance. On the brain level, PD is manifested by the death of dopaminergic neurons in the substantia nigra area, resulting in severe dopamine deficiency.

A 2019 study on mice has shown that developmental exposure to dieldrin, a toxic organochlorine pesticide used in the past as an alternative to DDT, can alter DNA methylation leading to an increased risk of PD [127]. In this study, pregnant female mice were fed food treated with dieldrin, and at 12 weeks after parturition, the level of DNA methylation in the ventral midbrain of the pups was measured. Authors found sex-specific DNA methylation alterations in pups exposed to dieldrin compared with controls, with a trend towards hypo- and hypermethylation in females and males, respectively. Importantly differentially methylated CpGs were found in Nr4a2 and Lmx1b genes, which are involved in the development of dopaminergic neurons, suggesting that the epigenetic remodeling caused by developmental dieldrin exposure could influence susceptibility to PD later in life [127].

Recently, Paul and coworkers showed that Parkinson’s disease can also be associated with exposure to Pb through epigenetic mechanisms [128].

Pb is a neurotoxic heavy metal that, when absorbed, is able to cross the blood-brain barrier and enter neurons and neuroglia. As a result, lead is capable of interfering with the release of neurotransmitters and energy metabolism and also causes the appearance of ROS.

The authors evaluated two large whole-blood DNAm datasets of PD patients and controls by applying a previously developed DNAm-based biomarker measuring lead exposure in the tibia or in the patella. The results of this analysis showed a positive association between PD status and the estimated chronic lead exposure in the tibia, indicating that DNAm changes in PD blood resemble those induced by lead exposure and supporting its contribution to PD pathogenesis [128].

Alzheimer’s disease (AD) is the most common neurodegenerative disease and is characterized by an accumulation of amyloid beta (Aβ) plaques, neurofibrillary tangles with the phosphorylated tau protein, and loss of synapses [129].

The association between the risk of developing AD and lead exposure is still under debate [130], but some interesting studies exist on this topic.

Bihaqi and colleagues [129] exposed rats to Pb at different life stages and showed that the AD-related genes beta amyloid precursor protein (AβPP), β-site amyloid precursor protein cleaving enzyme 1 (BACE1), and specificity protein 1 (Sp1) were upregulated in elderly rats exposed to the contaminant at an early age, resulting in cognitive impairment later in life. As these genes are rich in CpG sites, the authors suggested that their upregulation could be due to alterations in DNA methylation levels (loss of methylation) caused by exposure to the contaminant at an early age and maintained during the rest of life [129].

This study confirms the results of a previous work by Dosunmu and coworkers [131], which evaluated the association between global gene expression and DNA methylation in a case-control study on the cerebral neocortex of mice exposed to lead during development (from day 13 of gestation to day 20 postnatal) and unexposed mice (controls). Specifically, the mice were divided into 4 groups as follows: 3 young Pb-exposed mice (20 days), 3 young unexposed mice, 3 old Pb-exposed mice (700 days), and 3 old unexposed mice. At 700 days, researchers found 150 genes differentially expressed in Pb-exposed mice; deregulated genes were involved in immunity, metal-binding, metabolism, and transcription/transduction functions. Importantly, gene expression changes correlated with DNA methylation levels of the identified genomic regions [131].

Collectively, the above-described studies confirm that the developmental stage is a critical period for exposure to Pb and suggest that epigenetic alterations mediate the increased risk for the onset of neurodegenerative diseases occurring later in life.

Some studies suggest that the epigenetic remodeling caused by exposure to Pb can also mediate the risk for Autism Spectrum Disorders (ASDs), complex neurological development disorders characterized by difficulties in social relationships and the implementation of defined and repetitive behaviors [124,132,133,134]. In particular, it has been shown that Pb exposure of human embryonic stem cells during the neural rosette formation phase leads to a reduction in the expression of PAX6 and MSI1, genes that regulate proper brain development, and to DNA methylation remodeling of more than 1000 genomic regions, fundamental for neurodevelopment, calcium ion import, actin cytoskeleton and neuronal projections [124,134], all mechanisms that are also altered in ASD and result in aberrant neuronal connectivity patterns [124,135,136].

Alterations in gene expression caused by exposure to Pb during development were also confirmed in a study on zebrafish embryos. Pb reduces the expression level of dnmt3 and dnmt4, orthologues of human DNMT3b, leading to a reduction in global DNA methylation level. In addition, a reduction in DNMT1 activity and methylation rates following human exposure to Pb was also observed in the same study [137].

Pb is not the only environmental toxicant potentially linked to neurological diseases through epigenetic remodeling. In utero, exposure to BPA has been shown to result in reduced levels of Dnmt1 [81,138] and altered levels of Dnmt3a in the mouse brain [138]. It has also been noted that, in the brains of female mice, exposure to BPA causes an increase in the expression of solute carrier family 1 member 1 (SLC1A1), a gene implicated in terminating the postsynaptic action of the neurotransmitter glutamate and maintaining glutamate concentrations below neurotoxic levels. Thus, early exposure to BPA could alter SLC1A1 methylation levels during neural development and consequently cause sex-specific differences in behavior and social interactions in juvenile mice [138]. In addition, in the mouse hippocampus, early exposure to BPA caused alterations in the expression levels of the brain-derived neurotrophic factor (BDNF) gene, which is implicated in neuronal survival, stress response, and the onset of mood disorders [139].

Some recent evidence suggests that also air pollution can have an impactful effect on neurodegenerative diseases through DNA methylation. A study by Honkova and colleagues compared genome-wide DNA methylation in whole blood from Czech policemen working in three cities with different types of air pollution [140]. The authors found several CpG sites differentially methylated between the groups under investigation, partially mapping in genes related to neuronal functions, dopaminergic system, and neurodegenerative diseases [140]. These results suggest that air pollution is able to impact brain function and, in particular, to cause different effects depending on the type of atmospheric component, through its effect on DNA methylation [140].

## 7. Conclusions

In conclusion, in this review, we collected extensive evidence that exposure to various toxic agents has influenced the entire course of human evolution and remains detectable today in the genetic background of certain human populations. The contemporary time that we are living in represents a great challenge for our species due to the number of substances never encountered in human evolutionary history and because of the rapid changes in such exposomes. In this review, we reported biomedical and clinical studies (a summary is reported in Figure 3) and presented them from an evolutionary perspective. We showed that various toxic agents can have a high impact on human evolution in terms of biodiversity (genetic and epigenetic variation) and biological fitness. The central molecular mechanism underlying these processes is epigenetic variability and, in particular, DNA methylation, which indirectly may influence mutation rate and genetic variability itself. In conclusion, we focused on a central organ for our species, the brain, and reported how toxic agents are able to greatly influence DNA methylation in this tissue with consequences on neurological health outcomes.

## Figures and Tables

**Figure 1 biomedicines-10-03090-f001:**
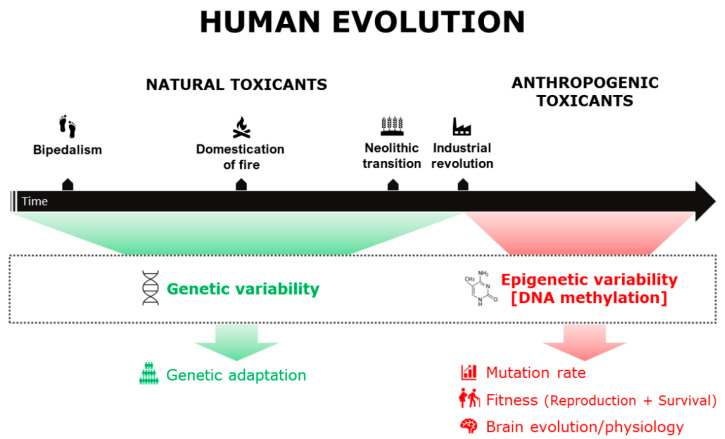
Illustrative timeline of the different events that exposed humans to different toxic substances during human evolution. Natural and anthropogenic toxicants, which in turn led to genetic adaptations and epigenetic variability, are reported in green and red, respectively. The effects of modern toxicants on DNA methylation played a key role in shaping the mutation rate, biological fitness (in terms of reproduction and survival), and evolution of the human brain.

**Figure 2 biomedicines-10-03090-f002:**
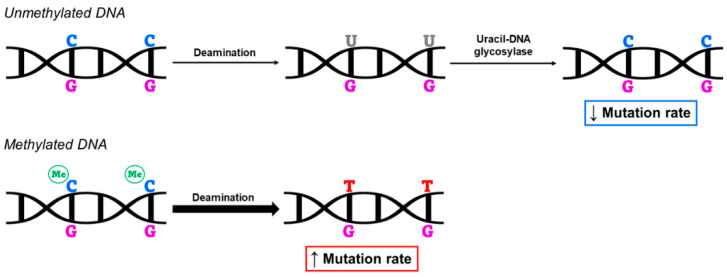
Schematic illustration of the difference between the deamination process of unmethylated cytosines, which are converted to uracil, and the deamination process of methylated cytosines, which are converted to thymine. The thicker arrow represents the higher conversion rate of methylated cytosines compared to unmethylated cytosines during the deamination process. The enzyme uracil-DNA glycosylase is able to eliminate uracil and promote DNA repair processes, while it is unable to eliminate thymine, causing the increased mutation rate. C = cytosine; G = guanine; U = uracil; T = thymine; Me = methyl group.

**Figure 3 biomedicines-10-03090-f003:**
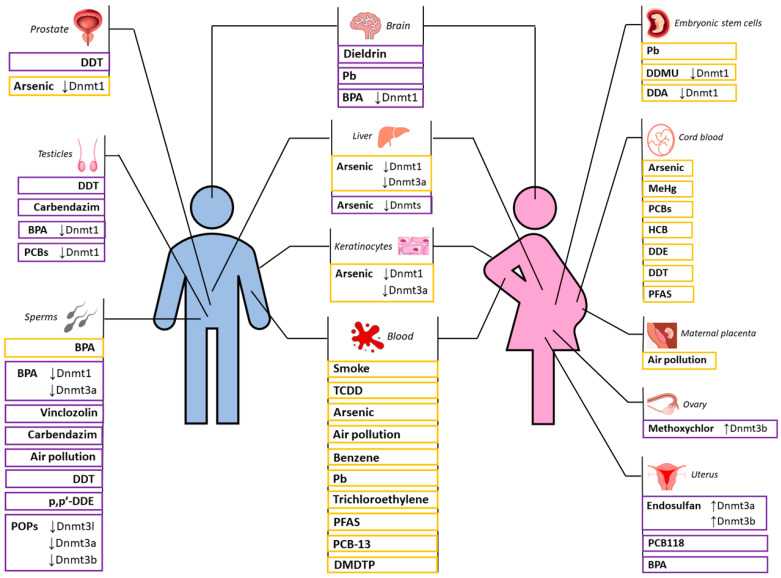
Schematic illustration summarizing the DNA methylation changes in different tissues after exposure to the main toxic agents examined in this review, with particular focus on the effect on Dnmts. The yellow boxes indicate the human studies, while the purple boxes indicate the mouse model studies.

**Table 1 biomedicines-10-03090-t001:** Summary of studies on the impact of different toxic substances on male reproductive fitness through DNA methylation.

Toxicant	Species	Tissue	Genes	Technology	Reference
DDT	rat	sperm	DMRs	MeDIP-chip	[72]
DDE	rat	sperm	IGF2	bisulfite sequencing	[73]
DDA DDMU	human	embryonic stem cells	DNMT1, Sox9, Oct4	HPLC-MS-MS	[74]
CBZ	mouse	leydig cells	5 mC	immunofluorescence staining	[76]
VCZ	mouse	sperm	H19, Gtl2, Peg1, Snrpn, Peg3	bisulfite-pyrosequencing	[77,78]
BPA	rat	testis	Er(α,β) DNMT3A	bisulfite sequencing methylation-specific PCR	[79] [80]
	mouse	sperm	whole genome	dot-blot assays	[82]
			DNMT1	RT-qPCR and western blot	
			mybph and prkcd	Affymetrix Mouse Promoter 1.0R Array	[83]
	human	sperm	LINE-1	methylation-specific real-time PCR	[84]
PCB118	mouse	testis	whole genome DNMT1, PCNA, STRA8	5 mC immunohistochemistry Western blot and qPCR	[85]
Heat stress	mouse	sperm	DMRs, Pik3cg, Nr4a1	bisulfite sequencing	[86]
POPs	rat	sperm	DNMT3L, DNMT3(A,B), Tbx2, Rapsn	RRBS and pyrosequencing	[87]
Mixture of MeHg, PCBs, HCB, DDE, DDT, PFAS	human	cord blood	whole genome	Infinium HumanMethylation450 BeadChip (Illumina)	[89]
Air pollution	mouse	sperm	whole genome	cytosine extension and methyl-acceptance assays	[90]
	human	sperm	whole genome	MethylFlash™ Global DNA Methylation (5 mC) ELISA Easy Kit (Epigentek)	[92]
Fire smoke	mouse	sperm	DMRs	RRBS	[91]

**Table 2 biomedicines-10-03090-t002:** Summary of studies on the impact of different toxic substances on female reproductive fitness through DNA methylation.

Toxicant	Species	Tissue	Genes	Technology	Reference
MXC	rat	ovaries	Erβ	BSPCR and MSPCR	[94,95]
			whole genome	methylation-sensitive AP-PCR	
			Dnmt3b	semiquantitative RT-PCR	
			PTEN, IGF-1, Rapid estrogen signaling	Nimblegen 3x720K CpG Island Plus RefSeq Promoter Arrays	[96]
Endosulfan	rat	utero	Erα, Hoxa10 Dnmt3(a,b) Erα	MSRE-PCR qRT-PCR immunohistochemistry and real-time RT-PCR	[98] [99]
PCB118	mouse	endometrium	Hoxa10, integrin subunit β, ER1	bisulfite genomic sequencing	[100]
BPA	mouse	uterus	Hoxa10	bisulfite sequencing	[101]
DDT DDE	rat	ovary, uterus, blood	-	-	[72,102]
oxidative stress (childbirth delay)	mouse	oocytes	whole genome, Dnmt3a, Dnmt3l	scBS-seq	[104,105]

## Data Availability

Not applicable.

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
