# Peer review of "Evolutionary Implications of Environmental Toxicant Exposure"

_biomedicines, 2022, doi:10.3390/biomedicines10123090_

Round 1
Reviewer 1 Report
The article as been reviewed carefully. The authors describe very important pathways with regard to epigenetics and evolution. However, the manucripts requires some modifications. Please note my specific comments below.
1)This reviewer highly recommends adding more figure to the review, for instance one showing the changes of de- and re-methylation during early embryonic development, and after genome activation.
2)Another figure should show the interaction betweeen DNMT1 and 3a and specific toxic agents, which are mentioned for instance in from L 535 on, and elsewhere.
3)In sections 4.1. and 4.2. dealing with the male and female fertility, the authors should also name heat stress, and oxidative stress as important stimuli of the environment inlfuencing fertiltiy. Furthermore, in section 4.2. the authors should mention in 5-10 sentences the effects of aging, and the current trend of postponing child bearing. Nowadays, people live longer than for instance in the 19th centuary, which is also an evolutionary effect with impacts on female fertiltiy, since more and more women desire to become pregnant in their late thirties.
Reviewer 2 Report
The manuscript titled “Evolutionary implications of environmental toxicants exposure” by Bolognesi et al. is an extensive review of how different types of toxic compound in the environment affects the different tissues and systems in the human body from the perspective of epigenetics, mostly DNA methylation. Although I understand that unavoidably summarizing such a large amount of work may result in a couple of pieces of inaccurate information, I do appreciate the authors’ efforts and found this manuscript interesting to both researchers from the same background and the general audience. I have a few suggestions which will hopefully help improve this manuscript:
1. Direct evidence on arsenic affecting DNMT/Tet proteins should be listed. I suggest including the DNMT part of the article below as a reference. https://www.ncbi.nlm.nih.gov/pmc/articles/PMC2877392/
2. Similarly, the discussion on Pb should include more direct evidence of how it affects DNA methylation. I suggest citing the following reference: https://pubmed.ncbi.nlm.nih.gov/27934997/
3. I suggest more figures in addition to Figure 1. It will help explain more. A potential figure could describe how the methylome of the different organs is affected by the different toxic compounds in schematics.
4. Additionally, it is recommended to add a table containing a list of toxic substances and their effect to the epigenome, as well as the relevant articles.
5. Citations [103] on epigenetic clocks should be expanded to include the following: doi 10.18632/aging.101508; doi 10.18632/aging.101414; doi 10.18632/aging.101684.
6. Minor point: Introduction missing on lines 42-43.
7. Minor point: In line 741 the word “epigenetic acceleration” does not make much sense. Maybe use “epigenetic age acceleration” instead.
8. Discussion on [112]: It is better to include which epigenetic clock found no difference, and discuss that there are multiple methylation clocks developed, sometimes giving different results.
9. Minor point: line 806 “and estimating also telomere length” It may be better to change the phrasing to “. Telomere length is also estimated from the DNAm data”.
Reviewer 3 Report
The authors reviewed the exposure to environmental toxicants that might involve human evolution from genetic and epigenetic variation. They tried to use many examples from previous literature to support the evolutionary implications of toxicant exposure. However, the manuscript was not well-organized and was not easy to read. I have two suggestions.
(1) The paragraphs in both “1. Introduction” and “4. Link between toxicants exposure, DNA methylation, and biological fitness “should be reorganized and combined for more clear expression. For example, the three paragraphs on lines 48-60 might be combined into one paragraph. There are some words lost on lines 42-43.
(2) Add some figures and Tables to help to understand. For example, a figure to explain DNA methylation influences mutation rate (lines 334-345); Tables for the toxicant exposure and biological fitness (sections 4.1 and 4.2).
Round 2
Reviewer 1 Report
No further comments, all suggestions have been addessed.
Author Response
Thank you for your comment!
Reviewer 3 Report
The manuscript has been improved, and I am impressed with the newly added figures and Tables. However, I have some minor suggestions.
1. Please use arsenic or As consistently.
2. What is meQTL on line 250? If it is a detecting method, the authors should describe it clearly.
3. Reference 33 is arsenic-related not diabetes mellitus. I suggest deleting the paragraph from lines 289 to 293 because it is not appropriate to mention disease-related DNA methylation in this section.
4. Too many paragraphs under a section, I suggest combining some paragraphs into one paragraph due to the same reference mentioned: lines 206-227; lines 228-241; lines 242-263, lines 264-181; lines 314-322; lines 323-332; lines 447-461 (please delete the cited page on lines 448-449); lines 521-532; lines 788-797.
5. Please change “paragraph” to “section” on line 788; delete “In this paragraph we mention” on line 790.
